# Neurons, Nose, and Neurodegenerative Diseases: Olfactory Function and Cognitive Impairment

**DOI:** 10.3390/ijms24032117

**Published:** 2023-01-20

**Authors:** Irene Fatuzzo, Giovanni Francesco Niccolini, Federica Zoccali, Luca Cavalcanti, Mario Giuseppe Bellizzi, Gabriele Riccardi, Marco de Vincentiis, Marco Fiore, Carla Petrella, Antonio Minni, Christian Barbato

**Affiliations:** 1Department of Sense Organs, Sapienza University of Rome, Viale del Policlinico 155, 00161 Roma, Italy; 2Institute of Biochemistry and Cell Biology (IBBC), National Research Council (CNR), Department of Sense Organs, Sapienza University of Rome, Viale del Policlinico 155, 00161 Roma, Italy; 3Division of Otolaryngology-Head and Neck Surgery, Ospedale San Camillo de Lellis, ASL Rieti-Sapienza University, Viale Kennedy, 02100 Rieti, Italy

**Keywords:** olfactory biomarkers, cognitive dysfunction, nasal neuroepithelium, neurodegenerative disease, neurons, nose, anosmia

## Abstract

Olfactory capacity declines with aging, but increasing evidence shows that smell dysfunction is one of the early signs of prodromal neurodegenerative diseases such as Alzheimer’s and Parkinson’s disease. The study of olfactory ability and its role in neurodegenerative diseases arouses much interest in the scientific community. In neurology, olfactory impairment is a potential early marker for the onset of neurodegenerative diseases, but the underlying mechanism is poorly understood. The loss of smell is considered a clinical sign of early-stage disease and a marker of the disease’s progression and cognitive impairment. Highlighting the importance of biological bases of smell and molecular pathways could be fundamental to improve neuroprotective and therapeutic strategies. We focused on the review articles and meta-analyses on olfactory and cognitive impairment. We depicted the neurobiology of olfaction and the most common olfactory tests in neurodegenerative diseases. In addition, we underlined the close relationship between the olfactory and cognitive deficit due to nasal neuroepithelium, which is a direct extension of the CNS in communication with the external environment. Neurons, Nose, and Neurodegenerative diseases highlights the role of olfactory dysfunction as a clinical marker for early stages of neurodegenerative diseases when it is associated with molecular, clinical, and neuropathological correlations.

## 1. Introduction

The olfactory system is involved in the detection and processing of odor signals. The processing of odors by sensory signals obtained from different chemical stimuli has a fundamental role in physiological and emotional homeostasis, playing a critical role in reproductive and neuroendocrine regulation. Olfactory detection begins in the nasal olfactory structures and proceeds to the first central station for olfactory processing, namely the olfactory bulb [1]. The olfactory bulb is an extension of the brain, representing the first sensory structure for direct contact with sensory stimuli and air environmental pathogens [2]. Experimental evidence suggests a concomitant role of olfactory bulb neurons in the onset of olfactory acuity decline and the onset of neurodegenerative pathology [3]. Olfactory stimuli are transduced through olfactory neuronal cell pathways before relaying the information to cortical brain structures for further processing [4].

### 1.1. Odor and Smell Impairment

Odor has an essential function in food, nutrition, the detection of environmental hazards, such as fires and gas leaks, and volatile chemicals [5,6,7]. Furthermore, it has been hypothesized that it is able to modify sexual behaviors and allows the determination of aspects of the person according to body odor [8]. Odor plays an important role in semantic memory, playing a role in reproduction for nutrition and neuroendocrine regulation. Smell is usually defined by several different abilities, such as olfactory threshold detection, identification, discrimination, and olfactory memory [9]. Quantitative olfactory performance can be classified into different ranges: normal or reduced (hyposmia) or absent (anosmia). Hyposmia and anosmia are estimated to affect 3–20% of the population [9]. Many factors influence smell, including physical genetic factors, nutrition, smoking, sex, head trauma, medical treatments, and exposure to viruses, because the nasal neuroepithelium is a direct appendix of the central nervous system in communication with the external environment without the protection of the blood–brain barrier (BBB) [10,11]. An impaired sense of smell negatively affects quality of life, food pleasure, and mood, affecting physical and mental well-being and social relationships [12,13]. Less than a quarter of people with olfactory disorders are aware of their problem until they are tested [10]. In addition, among adults aged 70 years and older, wrong identification rates for warning odors were 20% for smoke and 31% for natural gas [14], which is a major public health issue [15,16]. The risk of olfactory dysfunction increases with age and may be the result of acute and chronic sinonasal disease, upper respiratory infections, toxic chemicals, head injuries, and degenerative diseases [17,18,19].

### 1.2. Olfactory Dysfunction in Neurodegenerative Diseases

In the last decade, olfactory dysfunction was linked as an early sentinel sign of neurodegenerative disorders [20,21]. The high prevalence, early onset, and the development of fine olfactory tests have boosted interest in the search for olfactory dysfunction as an early marker of Parkinson’s disease (PD) and Alzheimer’s disease (AD) [22,23,24,25,26,27]. Olfactory impairment shows up in the early stages before the motor symptoms of PD [28]. The use of smell, as a biomarker for neurodegenerative diseases, is useful in the characterization of the prodrome stages, in identifying early diagnostic strategies and differential diagnoses, and in prediction of clinical cases. In general, the focus on olfactory function can help improve the chances of success of neuroprotective and disease-modifying therapeutic strategies [29]. Understanding the mechanisms related to olfactory function is essential to determine their association with neurodegenerative disorders. Specific anatomical systems and environmental factors can contribute to olfactory loss associated with neurological diseases, although the direct biological relationship with each disorder remains unsolved and further investigations are needed. Environmental stimuli that evoke pleasant familiar smells modulate the perception of stress and olfactory sensitivity, modifying the response to stress and the consequent physiological response during adulthood and aging [30,31,32]. In humans, the sense of smell has a significant influence on daily behaviors and quality of life. Impaired smell, associated with a decrease in olfactory ability, has been suggested as a preliminary indicator of classical neurodegenerative disorders such as AD and PD, now recognized as a prodromal neuropathology of the symptom associated with both circumstances.

### 1.3. Olfactory Biomarkers

In this manuscript, the inclusion and exclusion criteria were developed after the objectives of this research were finalized, but before any research was performed. The objectives of this review were to identify, delineate, and critically evaluate the link between smell and olfactory impairment as a clinical biomarker of early signs and symptoms of selected neurodegenerative diseases. We conducted a comprehensive literature analysis by searching the “PubMed” database. The principal search terms were ‘olfactory biomarkers’ and ‘cognitive dysfunction’. The filter criteria in the search strategy were publication date (recent papers published in the last ten years are more informative about a detailed olfactory test used in a clinical study), the language of publication (only looking at English full articles and not only abstracts), clinical studies performed in human subjects and not in animal models, olfactory test evaluation and not imaging/radiological scale, exclusion of many neurodegenerative diseases apart from PD, AD, MCI, and main psychiatric diseases, and exclusion of anosmia caused by COVID-19 infection and/or other comorbidity for olfactive impairment. The titles and abstracts of the identified papers were initially screened and selected by six independent reviewers (IF, GFN, FZ, LC, MGB, and GR) based on their relevance to the review topic. The literature search yielded 90 papers. From a critical recognition of works reporting these aims, more than 70 were excluded (Figure 1). 

At least eight studies were excluded because of the similarity in objectives, experimental paradigm, or final interpretation of the data reported by the researchers. With the intention of avoiding repetitions for readers, only original manuscripts were selected to analyze, and 12 studies were included and discussed (Table 1).

This review focused on olfactory impairment as a preclinical and early indicator of neurodegenerative disease and emphasizes the great potential of olfaction to be considered a clinical marker of these diseases and helpful in the characterization of prodromal stages of these diseases. We examine the literature with the aim to identify the molecular and cellular relationship between smell disorders and neurodegenerative diseases and consider the role of olfactory dysfunction as a screening parameter for their early diagnosis. 

## 2. Olfactory Pathways

Odor molecules entering the nasal cavity are first trapped in the mucus that lines the neuroepithelium of the olfactory fossa. Within the olfactory epithelium are neurons containing olfactory G-protein-associated receptors. Olfactory information is transmitted from the olfactory neurons in the nose via the olfactory bulb (OB) to several brain areas. Interneurons project to the primary olfactory cortex and anterior hippocampus, which proceed with smell recognition and memory retrieval. The hippocampus is involved in olfactory working memory, and odor identification involves Broca’s area and the orbitofrontal and prefrontal cortices, which are also included among the olfactory cortical regions. The piriform cortices, consisting of anterior and posterior sections, represent the structures where the first cortical processing of olfactory stimuli occurs. [33]. These olfactory structures are considered the primary olfactory cortex. Anterior olfactory nucleus cells are responsible for odor stimulation, and the piriform cortex plays an important role in processing odor signals based on experience and learning [33] (Figure 2).

### Olfactory Mucosa Molecular Mechanisms Neurobiology

The olfactory mucosa (OM) decorates the roof of the nasal cavity and ‘sniffs’ odorants as they enter it. The OM is a mucus-secreting tissue, structured by the olfactory epithelium (OE) and lamina propria (LP). The olfactory epithelium is composed of olfactory receptor neurons, also known as olfactory sensory neurons, bipolar neurons with dendritic filaments with protruding cilia projected into the mucosal cavity, and several types of supporting cells. Axons from olfactory receptor neurons synapse with secondary olfactory projection neuron dendrites, and the Mitral and tufted cells, forming a glomerulus [33,34,35]. The OE is composed of sustentacular cells, Bowman’s gland ductal cells, horizontal and globose basal stem cells, and microvilli cells. Sustentacular cells envelop the dendrites of the olfactory receptor neurons, tuning metabolic, secretory, and phagocytic functions.

The odorant molecule reaches the olfactory epithelium inside the olfactory cavity and binds to a specific odorant receptor, a protein belonging to the G protein-coupled receptor. The complex, odorant receptor/odorants, activates a cAMP-second messenger pathway and a specific olfactive G-protein, and activates downstream adenylyl cyclase III, increasing cAMP levels in the cilia processes.

Therefore, this triggers the Ca^2+^ influx, opening the Cl-channel, and induces the depolarization of the neuronal receptor axons propagating through the cribriform lamina, forming nerve fascicles reaching the glomeruli in the olfactory bulb.

In addition, the olfactory receptor neurons express one of the most numerous mammalian gene families, confirming the evolutionary importance and success of these genes. Each sensory neuron expresses one olfactory receptor, among 400 individuated. Each receptor binds several odorants, which can bind to several receptors. 

This apparently simple decoding network, resident in our noses, permits us to sniff thousands of odors, but the precise molecular mechanisms are unknown (Figure 2). An explanation of the molecular odorant transduction signals consists of conformational changes generated between odorant molecules and odorant receptors. Alternatively, the molecular binding of odorant receptors to odorous substances permits the electron transfer and G-protein cascade activation. Importantly, a large part of studies on olfactory cilia membranes, receptors, and G-protein binding regions were carried out in mouse models, reducing our molecular understanding of odorant transduction. Indeed, there are important anatomical differences between the vomeronasal organ and the terminal nerve, which connects the vomeronasal organ and the limbic system. In humans, they are vestigial organs, and the role of these receptors is debated.

## 3. Olfactory Tests for Smell Loss in Neurodegenerative Diseases

Olfaction ability can be tested using different behavioral olfactory identification tests to value odor detection threshold, odor identification, and odor discrimination [36]. Olfactory identification methods consist of the presentation of a suprathreshold concentration of an odor, and patients must make a choice from different items. Olfactory discrimination consists of differentiating odors, but not identifying these ones. Odor detections are measured by presenting different concentrations of a given odor, from the lowest concentration to the highest concentration, to identify the lowest odor concentration that can be perceived. Many tests (Table 2) have been used to quantify olfactory ability in neurodegenerative diseases, such as the University of Pennsylvania Smell Identification Test (UPSIT) and Sniffin Sticks Test (SST) [37,38]. UPSIT is a scratch-and-sniff microencapsulated odorant strip (40 items) that provides an indication of anosmia. After scratching with a pencil tip, the subject samples the smell and is asked to match it with one of four given choices. A response is required even when no odor is perceived. The test score consists of the total number of correctly identified items. The interpretation of a given subject’s test score is made by its comparison with age- and gender-matched population norms provided in the test manual. SST-12 or -16 olfactory screening tests, consisting of a smell identification pen-like, in which there are four option words depicting scent objects, in the form of a 4-min identification test, allowing for detection of anosmia and hyposmia [39,40]. It can also be used laterally, one nostril tested independently of the other. In addition, there is the Brief Smell Identification Test (B-SIT), which uses 12 test items derived from UPSIT, and the San Diego Odor Identification Test (SDOIT), in which there are common natural odors in opaque jars and options with pictures [41,42,43]. B-SIT is a quick, disposable screening test where the patient is asked to identify 12 odors contained in a microcapsule fixed on 12 strips of paper. This microcapsule is broken and scratched on with a pencil. The test is administered to the patient, who chooses from four possible answers to identify the smell. A score under 9 points suggests hyposmia, whereas a score under 4 corresponds to anosmia. The test is performed in 5 min, and both nasal nostrils are tested simultaneously.

Moreover, culturally specific tests have been developed, such as the Scandinavian Odor Identification Test (SOIT), Barcellona Smell test-24 (BAST24), the Odor Stick Identification Test for Japan (OSIT-J), and the Italian Olfactory Identification Test (IOIT) [44,45,46,47]. Another test was recently developed called Sniff Bubble, a novel olfactory threshold using rose odor-containing beads made with 2-phenyl ethyl alcohol (PEA), which is considered ideal for estimating olfactory acuity [48]. Furthermore, SCENTinel, a new rapid olfactory test developed by scientists and collaborators at Monell, is a tool that can help test for loss of sense of smell easily and self-sufficiently by the same patient [49,50]. Some odors are not universally recognized, and, for this reason, both the UPSIT and the Sniffin Sticks Test have been adapted and validated for use in many different languages and cultures, and normative values have been developed for age and gender (Table 2).

## 4. Loss of Smell and Aging

Environmental stimuli, aging, and cellular senescence contribute to the progress of a reduced response of olfactory structures [51]. During the ageing process, the number of olfactory receptor neurons decreases because the olfactory epithelium is gradually replaced by the respiratory epithelium; therefore, starting from 65 years of age, subjects can often manifest progressive hyposmia, up to anosmia [52]. Among the causes, the ossification of the cribriform plaque is associated with age and joined with the closure of the foramina, damaging the olfactory neuroepithelium and olfactory receptors that induce the loss of smell. Age-related loss of olfactory function is greater in males than females, as there is a gender difference in the total number of OB cells, highlighting that women have nearly 50% more OB cells than men, which could explain the sexual olfactory differences [53,54]. Event-related olfactory potentials also show greater latency with age, confirming that central structures are involved in the olfactory pathways. In fact, with ageing, there is a general decline in the ability to identify odors, which accelerates considerably above the age of 70. In older adults with no cognitive impairment, age is inversely related to odor identification test scores. In practice, this means that absolute odor test scores cannot be used to define abnormality and that an age adjustment must be used. Women score slightly better than men on odor identification tests under para-physiological conditions. Olfactory performance reaches its spike at 40 years, and, after, gradually drops with ageing [55]. Experimental evidence using magnetic resonance imaging (MRI) for healthy normal ageing individuals indicates a link between smell loss with a volume reduction and a morphological impairment of the olfactory nervous structures of the limbic system, such as the amygdala, entorhinal, and perirhinal cortex, and in olfactory-related regions of the cerebellum [56,57,58]. Different factors induce olfactory impairment in aging, such as nasal disease reduction of cribriform plate foramina and the reduction of olfactory receptor neurons. Age-related changes of smell function include the decrease of OB size due to the atrophy and loss of neural elements such as ORNs and olfactive epithelium damage. Age alterations effect, furthermore, the volume of the amygdala, piriform cortex, anterior olfactory nucleus, and hippocampus [59,60]. The decline of odor identification in normal older individuals has been linked with the increasing of cortical amyloid and neurofibrillary tangles in the entorhinal cortex and hippocampus, thanks to the fact that the olfactory capacity could be a valid biomarker of a good shape of the aging brain [61,62,63]. Furthermore, there is a connection between loss of smell and increased mortality risk in elderly people [64]. Minimal changes in olfactory perception reflect tunable olfactory dysfunction that anticipates several neurodegenerative diseases, and this may be determined by loss of synaptic function [64,65,66].

## 5. Smell and Cognitive Impairment in Neurodegenerative Diseases

Olfactory dysfunction is linked to many neurodegenerative diseases such as PD, mild cognitive impairments (MCI), and AD [67,68,69,70,71]. Olfactory disability precedes motor and cognitive symptoms, and, for that reason, it has been considered a prodromal symptom of neurodegenerative diseases. Several studies have confirmed that pathological protein aggregation seems to affect olfactory regions before other regions, suggesting that the olfactory system could be particularly vulnerable in neurodegenerative diseases [72]. Brain regions and olfactory function are connected by four associated proteins: α-synuclein, transactive response DNA-binding protein 43 (TDP-43), hyperphosphorylated tau, and β-amyloid proteins in olfactory neurons and mucosa. In AD patients, olfaction disability follows the accumulation of hyperphosphorylated tau and β-amyloid proteins in the olfactory system [73]. On the other hand, systematic olfactory measurement indicates a rapid olfactory decline during aging with normal cognition. Dementia and smaller grey matter volumes on 3T-magnetic resonance imaging indicates neurodegenerative processes identical in those with olfactory decline and AD. Olfactory impairment might be a potential biomarker for early AD detection [74] (Table 3).

### 5.1. Parkinson’s Disease

PD is a neurodegenerative disease with motor and non-motor symptoms, generally related to the modification of neurochemical pathways and anatomical structures. Establishing an early biomarker can allow for premature diagnosis and preventive treatments [75]. Olfactory dysfunction is a frequent and early non-motor clinical finding of PD, due to the primary involvement of both the peripheral olfactory system for sensing and the central olfactory structures required for identification and discrimination. Conversely, hyposmia has been reported to be approximately 30% less frequent in Leucine-rich repeat kinase 2 (LRRK2) patients than in idiopathic PD [76,77], while PD patient carriers of a mutation in the glucocerebrosidase gene seem to have impaired olfaction after the appearance of motor symptoms [78]. Recently, a comparative meta-analysis of over 1100 patients showed the olfactory performance of PD patients was worse than patients with other neurological disorders [79]. Up to 90% of preclinical PD cases showed an olfactory impairment preceding the onset of motor symptoms by decades [80]. Radiological imaging such as PET or functional MRI allows, respectively, for the study of the neurochemical pathways related to olfactory dysfunction and the identification of the structures of the central olfactory system involved in age-related olfactory regression as regions of the right amygdala and piriform cortex [81] in the pathological evaluation [82,83,84,85]. Multiple neurotransmitters are impaired in PD, and many have shown an association with olfactory loss, including, mainly, dopamine and acetylcholine. Among the various hypotheses, the most widely accepted is the association between the identification of abnormal odors and dopaminergic degeneration of the dopamine transporter [86]. The apoptosis of dopaminergic neurons and intra-neuronal α-synuclein aggregates, named Lewy bodies, are typical features of PD. The potential involvement of the olfactory area in PD was born by observation of Lewy bodies throughout the piriform cortex, the entorhinal cortex, and the cerebellum. Indeed, as in dementia with Lewy bodies, idiopathic rapid eye movement, sleep behavior, mild cognitive impairment, and AD, the accumulation of Lewy bodies in the olfactory bulb is directly related to the motor scores of the Unified Parkinson’s Disease Rating Scale, suggesting that the progression of the disease is related to the accumulation of the protein [87,88,89,90]. Physiological aging is also characterized by a progressive deactivation of the central structures, including the olfactory anatomical structures. The size of the olfactory bulb itself decreases, reflecting generalized atrophy caused by age [91]. Studies evaluating the activity and volume of brain structures showed differences in patients with PD compared to age-matched groups: in some studies, but not all, the volume of the olfactory bulb was reduced in PD [92,93]. It is evident that the early diagnosis of PD, as well as of AD, would allow the application of potential preventive therapeutic strategies, modifying their clinical course. It should also be considered that patients with PD and anosmia also showed abnormal structural integrity of central olfactory structures compared to patients with PD without olfactory dysfunction or age-matched healthy controls [94]. A leading role in this sense would perhaps be played by central cholinergic denervation, so much so that the effects are more evident in the limbic system, especially in the hippocampus. UPSIT scores and acetylcholinesterase activity correlates in patients with PD, confirming the role of the acetylcholine pathway in loss of smell.

### 5.2. Mild Cognitive Impairment

Olfactory dysfunction can be seen as a predictor for a conversion from MCI to AD. In the progression from cognitively normal to AD, patients often pass through a transition stage of mild cognitive impairment. MCI patients are not demented, and activities of daily living remain largely preserved. Dementia requires significant impairment and decline from a previous level of functioning that interferes with independence in activities of daily living in at least one of the following cognitive domains: learning and memory, language, complex attention, perceptual-motor function, or social cognition. Thus, prospective cohort studies confirm that olfactory deficit raises the risk to develop cognitive impairment. In fact, a strong increase of loss of smell predicts a conversion from MCI to AD [95,96]. Approximately 35% of patients with MCI transitioned to dementia within one year. Other studies have indicated conversion rates of 15% to 24% within two years, or approximately one-third over three years [96]. A three-year follow-up showed a combination of olfactory decline, verbal memory, hippocampus volume, and entorhinal cortex volume. In addition to UPSIT scores, measures of episodic verbal memory, informant reports of functional decline, and magnetic resonance imaging of hippocampal and entorhinal cortex atrophy had a strong value (90% specificity and 85% sensitivity) to predict accuracy for the transition from MCI to AD varying across studies [97,98]. Typically, MCI is assessed with neuropsychological testing such as the Mini-Mental Status Examination. Advancements in the ability to predict progression from cognitively normal to MCI and MCI to dementia will be essential for clinical trial design and for patients to make treatment decisions. [99,100]. Based on our current knowledge, the olfactory test should be in assessment for the systematic screening of subclinical AD.

### 5.3. Alzheimer’s Disease

AD is a widely neurodegenerative disease characterized by a progressive decline in memory functions, progressive dementia, trait personality changes, and a reduction of language functions. It is estimated to affect more than 50 million people worldwide [101,102]. AD has an incidence of 1% in subjects aged from 60 to 70 years. It will continue to increase with the population longevity, with a projection to increase to 13.8 million by 2060 in the USA. The precise mechanism behind the olfactory impairment in AD is yet unknown, but AD patients have impaired olfaction joined with an accumulation of hyperphosphorylated tau and β-amyloid proteins in the olfactory system [103,104,105,106]. Olfactory dysfunction in AD was reported nearly fifty years ago in 1974 by Waldton [106]. During this decade, experimental works evaluated that MCI patients developed AD, showing a measurable olfactory impairment [107,108,109]. Sniffing tests contribute to individuate preclinical symptoms of AD, predicting that the olfactory impairment is linked to early phases of AD [110,111], and they are associated with the level of Aβ 1-42. The loss of olfactory identification is correlated with many markers of neurodegeneration such as the reduction of the entorhinal cortex, hippocampus, and amygdala volumes. In addition, the decrease in left hippocampal volume has been linked with a disruption in odor recognition ability in AD patients [112]. Moreover, using neuroimaging, it has been verified that there is a volumetric decrease in the olfactory bulb (OB) and primary olfactory cortex (POC) associated with the clinical diagnosis of AD. Furthermore, fMRI has shown that blood oxygenation level signal in the POC was weaker in AD patients than in healthy people [108,113,114,115]. Deficiency in odor identification is associated with a sensitivity and specificity of about 87–91% in AD versus normal cognition diagnosis [116]. This approach shows confidence with respect to cerebrospinal fluid biomarkers, but reduced predictability to amyloid imaging and MRI [117]. On the other hand, the ratios of cerebrospinal fluid (CSF) and total tau and phospho-181-tau compared to Aβ 1-42 were increased. One of the early brain regions affected is the olfactory system, which has shown in postmortem studies the presence of amyloid beta plaques (Aβ) and neurofibrillary tangles (NFTs) on the OB, correlated with the severity of AD [117]. Importantly, the areas targeted early in AD tau pathology are also areas important for processing olfactory information and olfactory dysfunction, such as odor identification, discrimination, and detection threshold tasks, as an early symptom of tau pathology [118,119,120]. AD patients have more problems with odor discrimination ability than odor identification because olfactory discrimination requires a more complex neurological processing than odor identification [121]. Neurofibrillary tangles and plaques appear very early in the olfactory bulb and anterior olfactory nucleus. Afferent neurons bring signals from the olfactory bulb into brain regions including the entorhinal cortex, piriform cortex, and amygdala. Neuropathology in the entorhinal cortex could disconnect incoming olfactory information from the hippocampus and disrupt the normal olfactory process, and neuroimaging shows the degeneration of the primary olfactory cortex, entorhinal cortex, and hippocampus in AD patients [122]. The olfactory tests sustain the clinical differential diagnosis of AD, and they could be more related to Aβ increasing. The exact mechanism that links olfactory dysfunction to biomarkers and clinical early stages of AD is undisclosed, and more effort is needed in this direction.

### 5.4. Olfactory Function in Neuropsychiatric Disorders

Olfactory impairment and low scores on the olfactory identification scale were associated with neuropsychiatric disorders and Major Depressive Disorders (MDD) [123,124,125]. A correlation emerged as patients diagnosed with depression show reduced olfactory performance, while patients with olfactory dysfunction show worsening depressive symptoms, which are relatively more acute in anosmic than hyposmic subjects [126]. Patients with MDD have a negative impact on primary and secondary olfactory cortical areas. Indeed, these patients have a decline in the activation of the thalamus, insula, and left middle orbitofrontal area, which represent the secondary olfactory areas. In addition, there is a reduction in nasal pathology of the olfactory performance of the piriform cortex, which is the primary region of the olfactory pathway [127]. Patients with a diagnosis of post-traumatic stress disorder and MDD show an olfactory impairment due to the anatomical thigh connection from the olfactory system with brain circuits that are involved in memory and cognition, and it contributes to a functional alteration in MDD individuals. With the aim to study the olfactory dysfunction by a comparison between patients with frontotemporal dementia, depression, schizophrenia, and bipolar disorder, it was evidenced that frontotemporal dementia patients showed severe impairments in olfactory identification, but not in odor discrimination [128].

## 6. Conclusions and Future Directions

Pathologic profiles of olfactory function may be based on differences in the neurocognitive or neuroanatomical processes that drive odor identification and discrimination. To identify odors in odor identification tasks, higher cognitive skills are required, including verbal and semantic skills [129,130]. Semantic memory, verbal capacity, and odor identification are correlated with the grey matter volume of brain regions from the right insular cortex to the right superior temporal gyrus [130]. In fact, AD is characterized by grey matter reduction in the right entorhinal cortex and right para-hippocampal gyrus [131], resulting in odor identification impairments. Similarly, studies on schizophrenia patients evidenced that they have a reduced posterior nasal cavity and olfactory bulb volumes, as well as smaller depth of the olfactory sulcus [132], a characteristic founded in the neurodegenerative course of AD patients [133,134]. Considering several common neuroanatomical defects in cognitive areas, in psychiatric and neurological chronic diseases, it is conceivable that neurodevelopmental alteration that involves olfactory routes could be a common trait of neurocognitive profile based on neuronal olfactory physiology. Including aging, genetic variability in odorant receptors, different cognitive capacities, and processing to discriminate the odors, different technical modalities of olfactory tests are all tunable elements to consider, establishing whether patients have decreased olfactory function associated with neurodegenerative diseases. As mentioned above, it is very difficult to obtain a sharp separation of healthy age-related smell loss from neurodegenerative disorders based on smell testing only, leaving an open question. During aging, the olfactory epithelium, containing olfactory receptors and axons reaching the olfactory bulb, is gradually being replaced by respiratory epithelium. Consequently, the activation of olfactory receptors by volatile molecules is reduced in aging versus youth, accounting for the different score values obtained in the olfactory tests, similarly to neurodegenerative diseases, frequently associated with older age groups. A partial solution to this observation was exploited by using an odor mixture of several molecules, increasing the probability of obtaining an adequate receptor response [135]. Odor mixtures have a higher probability to be measured with respect to single molecules, and, in addition, this combination could discriminate the involvement of central olfactory brain regions, consisting of low scores in both conditions with respect to a decreased sensitivity measured by single molecules potentially associated with a specific pathway or neuronal route [136]. After sniffing light molecules (<150 g/mol) and heavy molecules (>150 g/mol), older people were less sensitive to heavy molecules with respect to young adults. Conversely, the olfactory training was associated with olfactory improvement [137]. At least, an important aspect for clinicians is represented by patients’ collaboration in the executive phase of the olfactory test. In this context, subjects suffering from neurodegenerative diseases should be able to perform tests that are not cognitively complex, in the short term, and few tasks. Since this objective is limited, the neuroimaging could integrate olfactory tests, but fMRI is highly specific and expensive as routine investigation [138]. For this purpose, the support of otolaryngologists is recommended, because they can perform a differential diagnosis for common nasal pathology frequently associated with anosmia or other olfactory impairment (Figure 3).

Our sense of smell is fundamental to many aspects of life and health. From an evolutionary view, the nose is crucial for alerting us to immediate or potential danger, nutrition, social relationships, and sexual attraction. Conversely, olfactory impairment is associated with disorders and pathologies and is usually the first sign of deterioration in neurodegenerative diseases, such as AD [139] PD [140], frontotemporal dementia [128], MCI [100], and psychiatric disorders such as MDD [124], schizophrenia [128], and bipolar disorders [141]. Importantly, changes in olfactory function often occur years before the clinical phase of these disorders and are, therefore, considered potential clinical markers and markers of disease progression [103]. In conclusion, it is possible to presume that (i) epidemiological studies underlined that the prevalence and the severity of the loss of smell grow with aging, even though the mechanisms below the olfactory dysfunction are yet unclear; (ii) there is evidence that olfactory dysfunction in cognitively normal persons may be interpreted as a biomarker of preclinical neurodegenerative disease; (iii) olfactory decline often precedes cognitive and motor symptoms as a prodromal manifestation of some diseases as well as PD and AD; (iv) the olfactory decline in association with other clinical signs and clinical symptoms and neuroimaging markers could be used for a preclinical diagnosis and therapeutic strategies; (v) despite several neuropathological mechanisms that have been related to the decline of olfactory function in neurodegenerative diseases, clinical and preclinical research on cellular and molecular mechanisms underlying the olfactory impairment is required. In this view, the integration of clinical studies on smell deterioration joined with gene expression profiles [103], and alternative therapeutic approaches, such as olfactory stimulation [142], might disclose the use of olfactory impairment, such as hyposmia and anosmia, as a sensitive biomarker in clinical pictures of neurodegenerative diseases. Altogether, these findings suggest that olfactory impairment correlates with cognitive performance, and that the olfactory tests are a valuable tool to predict the risk of dementia [143]. The connection of olfactory and cognitive decline resides in the nasal neuroepithelium, which contains neuronal cells bridging the external environment with the central nervous system without the protection of the blood–brain barrier. The nasal neuroepithelium has become an emerging anatomical area under investigation after the SARS-CoV2 pandemic [144,145]. Ongoing studies on COVID-19 anosmia [146] could reveal new molecular aspects unexplored in olfactory impairments due to neurodegenerative diseases, shedding a light on the validity of smell test predictivity of cognitive dementia. The neuroepithelium might become a new translational research target (Neurons, Nose, and Neurodegenerative diseases) to investigate alternative approaches for intranasal therapy and the treatment of brain disorders. 

## Figures and Tables

**Figure 1 ijms-24-02117-f001:**
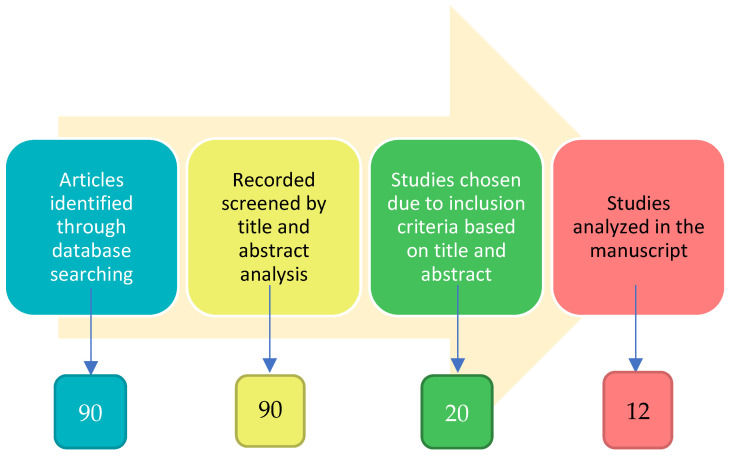
Flow chart of articles selection (see methods).

**Figure 2 ijms-24-02117-f002:**
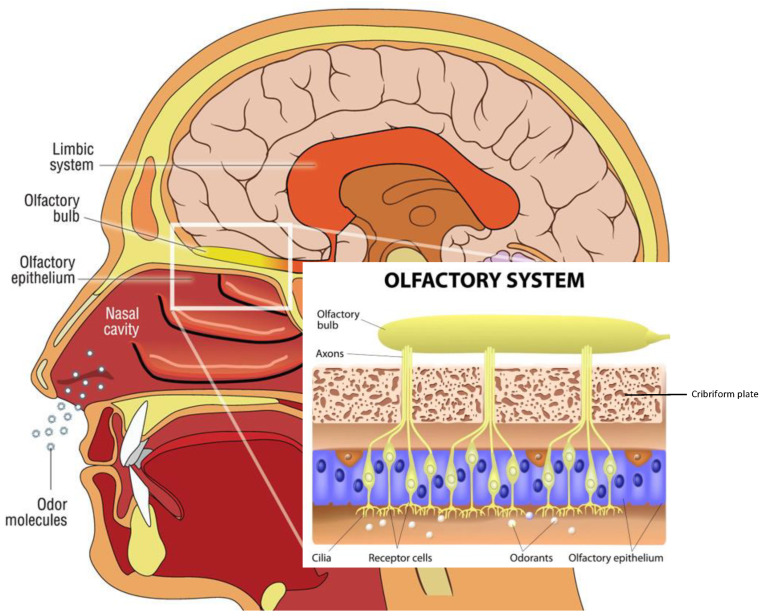
Descriptive anatomy of odorant signal transmission and the human olfactory system. In the box, a focus is on the olfactory system, with neuroepithelium, composed of olfactory sensory neurons that, through the lamina cribrosa plate, establish a synaptic connection with the mitral cells of the olfactory bulb. The axons of the monosynaptic mitral cells that make up the olfactory tract bifurcate at the terminus, fornix, or olfactory cortices. Primarily, the limbic system with the pyriform cortex, amygdala, and entorhinal cortex is involved. (Modified from medical illustration by Patrick J. Lynch).

**Figure 3 ijms-24-02117-f003:**
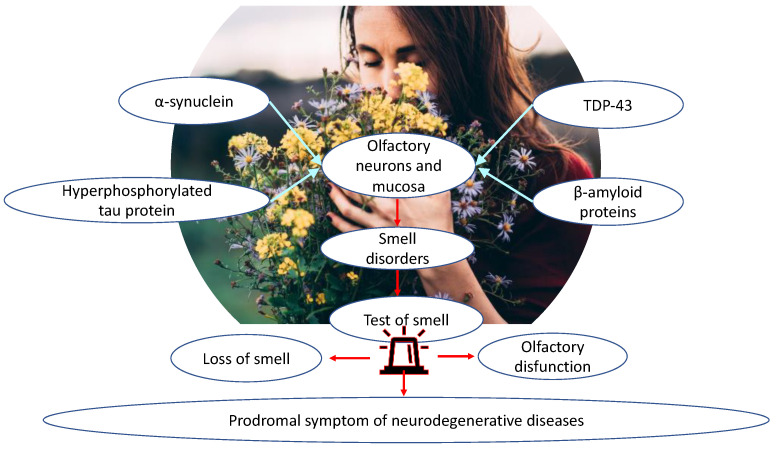
Higher cognitive abilities, reserved for specific brain regions, are required to identify odors, which, in neurodegenerative diseases, manifest common neuroanatomical defects, up to altered neurotransmission patterns of olfactory pathways, such as for the deposition and accumulation of molecules such as α-synuclein, TDP-43, hyperphosphorylated tau protein, and β-amyloid protein. An important aspect for the clinician, and particularly the otolaryngologist, is the performance of olfactory testing in patients who manifest olfactory disorders. In this context, individuals affected by neurodegenerative diseases should be able to perform non-cognitively complex, short-term tests with few tasks. For this reason, differential diagnosis is recommended for common nasal disorders frequently associated with anosmia or other olfactory disorders so that such disorders can be identified early, as a prodromal symptom of neurodegenerative diseases.

**Table 1 ijms-24-02117-t001:** Main studies discussed in the review (see text for description).

Year	Country	Study	Purpose
2013	USA	Observational Study	Outline how progressive changes in olfaction may be used as a biomarker of cholinergic denervation and cognitive decline in Parkinson’s Disease patients
2014	UK	Review	Review of potential biomarkers of prodromal Dementia with Lewy bodies
2016	USA	Observational Study	Standardized tests of odor identification ability may provide a useful tool to improve diagnostic and predictive accuracy for cognitive decline
2017	USA	Review	Potential for olfaction as a biomarker for early or differential diagnosis and prognosis in Parkinson’s Disease
2018	Spain	Review	Olfactory function measurements in neurodegenerative diseases
2018	China	Review	Review of studies about markers of imaging and neurophysiological, genetic, cognitive, autonomic function of Rapid Eye Movement and their predictive value for neurodegenerative diseases
2019	Germany; USA	Review	Identification of new marker for Prodromal Parkinson’s Disease
2020	Canada	Meta Analysis	Verify whether the presence of Subjective Cognitive Decline is associated with a decrease in olfactory identification ability
2020	Austria; Germany; France	Review	Assemble current knowledge from different medical fields on olfactory/gustatory dysfunction
2022	China	Review	Review of the olfactory evaluation of Alzheimer’s Disease model mice
2022	UK	Review	Review of the recent research from longitudinal research studies in isolated Rapid Eye Movement
2022	Canada	Review	Review of the recent research from translational research studies in Drug Delivery N2B Neurological and Psychiatric Illnesses

**Table 2 ijms-24-02117-t002:** Main tests for olfactory function and their clinical application.

Test	Type	Response Mode	Parameters Test
University of Pennsylvania Smell Identification Test (UPSIT)	Scratch-and-sniff micro-encapsulated odorant strips (40 items)	Four option words for each odorant	Identification
Sniffin’ Sticks Test (SST)	12 or 16 smell identifications pen-like	Four option words depicting scent object	Threshold, identification, discrimination
Brief Smell Identification Test (B-SIT)	12 test items derived from UPSIT	Four response alternative words	Identification
San Diego Odor Identification Test (SDOIT)	Common natural odors in opaque jars	Options with pictures	Identification
Scandinavian Odor Identification Test (SOIT)	Odors culturally validated by Scandinavian people	Four alternatives	Identification
Barcellona Smell test-24 (BAST24)	24 odors scoring test detection	Four option words for each odorant	Forced choice Smell detection, Identification
Odor Stick Identification Test for Japan (OSIT-J)	13 odors scoring test	Four option words for each odorant	Smell detection
The Italian Olfactory Identification Test (IOIT)	33 micro-encapsulated odorants	Four possible answers	Identification
Visual Analog Scale (VAS)	10-cm line, both ends of which have statements of the maximal and minimal extremes	Marking the line at the appropriate point between the two extreme statements, defined as “anosmia” and “normal”	Identification
Sniff Bubble Rosy Smell	PEA, which has Rosy smell, from 1, highest concentration, to 7, lowest concentration	Score from minimum to maximum (anosmia-normal)	Identification
SCENTinel 1.0	Flower odor (Givaudan; perfume compound with 2-phenylethanol [CAS No. 60-12-8] as the main components	“First attempt” is a four-alternative forced choice. “Second attempt” is a three-alternative forced choice, intensity range: 1–100	Odor detection, intensity, and identification

**Table 3 ijms-24-02117-t003:** Neuropathological processes of the olfactory system in neurodegenerative diseases. Gross anatomy of the nose, the olfactory nerve, and the olfactory bulb, and cellular and molecular neuropathology. (OSN: olfactory sensory neurons, DA: dopaminergic, OT: olfactory tract, PC: piriform cortex, AMG: cortical nucleus of the amygdala, EC: entorhinal cortex, OFC: orbitofrontal cortex.)

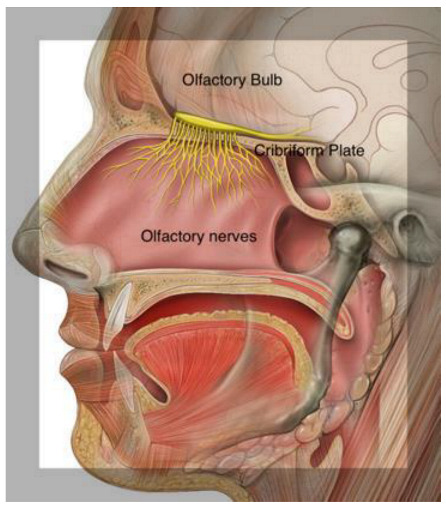	Neuropathology hallmarks *Olfactory mucosa:* β-amyloid aggregates OSN degeneration Occlusion of the foramina α-synuclein aggregates *Olfactory bulb:* Reduction in bulb volume Tauopathy Diffused β-amyloid aggregates α-synuclein aggregates Increase in DA neurons Axonal loss in OT NFTs and core plaques *Olfactory cortices:* Tauopathy and α-Synucleinopathy in: OT AMG PC EC OFC

## Data Availability

Not applicable.

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
