# Peer review of "Neurons, Nose, and Neurodegenerative Diseases: Olfactory Function and Cognitive Impairment"

_ijms, 2023, doi:10.3390/ijms24032117_

Round 1

Reviewer 1 Report

The review manuscript "Neurons, Nose and Neurodegenerative Diseases: Olfactory Function and Cognitive Impairment." has been prepared and is well presented.  As a clinical marker of neurodegenerative disorders, olfactory dysfunction was reviewed in the study. An alarming rise in neurodegenerative prevalence necessitates such a review around the globe.

 There are a few minor improvements I would like the authors to make to their manuscripts

1.     Throughout the manuscript, some of the words are not abbreviated, and even when they are, they are not used wisely.

2.     Section 5, “Pathophysiology of olfaction and cognitive impairment in neurodegenerative diseases” should be detailed to improve the manuscript.

3.     A single paragraph introduces the manuscript. It can be truncated into short paragraphs with subheadings. The reader might not be interested in a long paragraph and it may be quite boring.

4.     Introduction Page 2 – Line No. 77, Sentence starting with “ Environmental stimuli ,,,,,,,, physiological impact is not clear. It can be rephrased.

5.     Section  2 Olfactory pathways – 119 to 121, The role of each part of the brain and piriform cortex, if it elaborated it might sound good.

6.     Section  2.1 Olfactory mucosa and molecular chemosensory mechanism. The molecular mechanism is not clear. Consider revising the chemosensory mechanism.

7.     Section  3- Line no. 171 Before UPSIT, the word "test" might not be needed.

8.     Section 4 - Loss of Smell and ageing, Line 190, “Several factors,,,,,,,,,,, activation”. Unable to understand clearly.

9.     Section 4 - Line 209 – Magnetic Resonance Imaging, popularly called MRI has not been abbreviated.

10.  Section 4 - Line no. 215- Both abbreviated and expanded forms are written. It can be avoided

11.  Table 3, In the legend the word has been abbreviated, and in the
table, it is in expanded form. Can be avoided to make the manuscript precise.

12.  Section 5.1 – Line 255, LRRK-2 PD can be expanded (Leucine-rich repeat kinase 2 (LRRK2)

13.  Section 5.1 – Line 261 fMRI can be expanded and positron emission tomography can be abbreviated.

14. In Lines 334, 338, 341, and 354 the words POC, CSF, NFT, and Ab can be expanded.

Author Response

Dear Editor,

We are appreciative of the comments and suggestions made by reviewers and believe that addressing their comments and concerns substantially improved our manuscript. All the changes in the manuscripts have been highlighted (red text). Please see below for point-by-point responses.  As suggested, the manuscript was checked by a native English-speaking colleague.

Reviewer 1) Comments and Suggestions for Authors

The review manuscript "Neurons, Nose and Neurodegenerative Diseases: Olfactory Function and Cognitive Impairment." has been prepared and is well presented.  As a clinical marker of neurodegenerative disorders, olfactory dysfunction was reviewed in the study. An alarming rise in neurodegenerative prevalence necessitates such a review around the globe.

There are a few minor improvements I would like the authors to make to their manuscripts

  1. Throughout the manuscript, some of the words are not abbreviated, and even when they are, they are not used wisely.

R1) as suggested, we corrected some of the words not abbreviated, throughout the manuscript.

  1. Section 5, “Pathophysiology of olfaction and cognitive impairment in neurodegenerative diseases” should be detailed to improve the manuscript.

R2) The section 5 was entitled ‘Smell and cognitive impairment in neurodegenerative diseases’, reflecting more appropriately the relative sub-paraghraphs.

  1. A single paragraph introduces the manuscript. It can be truncated into short paragraphs with subheadings. The reader might not be interested in a long paragraph and it may be quite boring.

    R3) as suggested, we reorganized the introduction into short paragraphs as follow:

1.1. Odor and smell impairment; 1.2. Olfactory dysfunction in neurodegenerative diseases ; 1.3. Olfactory biomarkers.

  1. Introduction Page 2 – Line No. 77, Sentence starting with “ Environmental stimuli ,,,,,,,, physiological impact is not clear. It can be rephrased.

R4) The sentence was replaced by: ‘Environmental stimuli that evoke pleasant familiar smells modulate the perception of stress, and olfactory sensitivity, modifying the response to stress and the consequent physiological response, during adulthood and aging’.

  1. Section  2 Olfactory pathways – 119 to 121, The role of each part of the brain and piriform cortex, if it elaborated it might sound good.

R5) The sentence was elaborated indicating the brain area functions: “Olfactory information is transmitted from the olfactory neurons in the nose via the olfactory bulb (OB) to several brain areas. Interneurons project to the primary olfactory cortex and anterior hippocampus which proceed with smell recognition and memory retrieval. The hippocampus is involved in olfactory working memory, and odor identification involves Broca's area and the orbitofrontal and prefrontal cortices, which are also included among the olfactory cortical regions. The piriform cortices, consisting of anterior and posterior sections, represent the structures where the first cortical processing of olfactory stimuli occurs.”

  1. Section  2.1 Olfactory mucosa and molecular chemosensory mechanism. The molecular mechanism is not clear. Consider revising the chemosensory mechanism.

R6) As requested, the section 2.1 was revised and reorganized as follow: “The odorant molecule reaches the olfactory epithelium inside the olfactory cavity, binds to a specific odorant receptor, a protein belongs to the G protein-coupled receptor. The complex, odorant receptor/odorants, activates a cAMP- second messenger pathway and a specific olfactive G-protein, activates downstream adenylyl cyclase III, increasing cAMP levels in the cilia processes. As consequence, triggers the Ca2+ influx opening Cl- channel, induces the depolarization of the neuronal receptor axons, propagating through the cribriform lamina, forming nerve fascicles reaching the glomeruli in the olfactory bulb.”

  1. Section  3- Line no. 171 Before UPSIT, the word "test" might not be needed.

R7) the word "test" was deleted

  1. Section 4 - Loss of Smell and ageing, Line 190, “Several factors,,,,,,,,,,, activation”. Unable to understand clearly.

R8) the sentence was replaced with: “Several factors influence olfactory processing, and molecular pathways associated with such activation”.

  1. Section 4 - Line 209 – Magnetic Resonance Imaging, popularly called MRI has not been abbreviated.

R9) abbreviation (MRI) was added

  1. Section 4 - Line no. 215- Both abbreviated and expanded forms are written. It can be avoided

    R10) the expanded forms were deleted.
  2. Table 3, In the legend the word has been abbreviated, and in the
    table, it is in expanded form. Can be avoided to make the manuscript precise.

R11) As suggested, the words were abbreviated in the table.

  1. Section 5.1 – Line 255, LRRK-2 PD can be expanded (Leucine-rich repeat kinase 2 (LRRK2)

R12) The abbreviation LRRK-2 was expanded

  1. Section 5.1 – Line 261 fMRI can be expanded and positron emission tomography can be abbreviated.
    R13) The abbreviation fMRI was expanded and abbreviation PET added.
  2. In Lines 334, 338, 341, and 354 the words POC, CSF, NFT, and Ab can be expanded.
  3. R14) The abbreviations POC, CSF, NFT, and Ab were expanded.

Thank you for appreciating our work and for helping us to make it clearer to the reader with these suggestions.

Reviewer 2 Report

A clear and well written manuscript.

Author Response

Dear Editor,

We are appreciative of the comments and suggestions made by reviewers and believe that addressing their comments and concerns substantially improved our manuscript. All the changes in the manuscripts have been highlighted (red text). Please see below for point-by-point responses.  As suggested, the manuscript was checked by a native English-speaking colleague.

Reviewer 2) Comments and Suggestions for Authors

A clear and well written manuscript.

We thank you again, for your contribution to improving our work and for your appreciation.

Reviewer 3 Report

This literature review focuses on studies assessing the early loss of smell and its association with neurocognitive diseases. This is a novel and interesting topic to review that is relevant to the aging population and the increasing incidence of neurocognitive diseases. However, the grammar and wording makes the understanding of the content and conclusions of this review hard to follow. That being said, the following issues should be addressed:

#1 The grammar and wording should be revised. The meaning of some sentences is unclear because of the vague and confusing wording. For instance, lines 43–44, 46–49, and 58–60 are sentences that are unclear.

#2 When discussing the papers that were selected based on the inclusion and exclusion criteria, it is stated that 90 papers were originally found and 74 were excluded. It is then stated that only 12 studies were used in this review. However, there is a difference of 4 papers that are unaccounted for. Please clarify what happened to these extra papers. Similarly, the flow chart in Figure 1 has different numbers (90, 90, 20, and 12). Please elaborate on how the 20 papers were originally chosen and how eight were excluded in the final analysis.

#3 In Figure 1, it is unclear what “qualitative synthesis” means. Please clarify.

#4 On lines 172–174, the explanations of the UPSIT, SST, and B-SIT is unclear and confusing. Please clarify and elaborate on what each test is comprised of and how they are used.

#5 On lines 184–187, it is stated that “…the UPSIT and the Sniffin’ Sticks Test have been adapted and validated for use in many languages and cultures…” and they have been normalized for different ages and genders. However, it is also stated that “…for that reason other tests have been introduced.” If the UPSIT and Sniffin’ Sticks Test have been modified for such a broad usage, why were other tests developed and why would the seemingly universal and normalized usage of the UPSIT and Sniffin’ Sticks Test be a reason to introduce even more tests? This statement is unclear.

#6 On line 234, please clarify what “They” refers to. 

Author Response

Dear Editor,

We are appreciative of the comments and suggestions made by reviewers and believe that addressing their comments and concerns substantially improved our manuscript. All the changes in the manuscripts have been highlighted (red text). Please see below for point-by-point responses.  As suggested, the manuscript was checked by a native English-speaking colleague.

Reviewer 3) Comments and Suggestions for Authors

This literature review focuses on studies assessing the early loss of smell and its association with neurocognitive diseases. This is a novel and interesting topic to review that is relevant to the aging population and the increasing incidence of neurocognitive diseases. However, the grammar and wording makes the understanding of the content and conclusions of this review hard to follow. That being said, the following issues should be addressed:

#1 The grammar and wording should be revised. The meaning of some sentences is unclear because of the vague and confusing wording. For instance, lines 43–44, 46–49, and 58–60 are sentences that are unclear.

#R1 Grammar and wording have been submitted extensive revisions in English.

-The sentence (lines 43–44) was deleted and replaced with:

“Olfactory stimuli are transduced through olfactory neuronal cell pathways before relaying the information to cortical brain structures for further processing [4].”

-The sentence (lines 46–49) was deleted and replaced with:

Odor has an essential function in food, nutrition, the detection of environmental hazards, such as fires and gas leaks, and volatile chemicals [5,6,7,]. Furthermore, it has been hypothesized that it is able to modify sexual behaviors and allow to determine aspects of the person according to body odor [8].

-The sentence (lines 58–60) was deleted and replaced with:

The impaired sense of smell negatively affects the quality of life, food pleasure, and mood, affecting physical and mental well-being and social relationships [12,13].

#2 When discussing the papers that were selected based on the inclusion and exclusion criteria, it is stated that 90 papers were originally found and 74 were excluded. It is then stated that only 12 studies were used in this review. However, there is a difference of 4 papers that are unaccounted for. Please clarify what happened to these extra papers. Similarly, the flow chart in Figure 1 has different numbers (90, 90, 20, and 12). Please elaborate on how the 20 papers were originally chosen and how eight were excluded in the final analysis.

#R2 Thank you again for your suggestions. The sentences were corrected and replaced with:The literature search yielded 90 papers. Subsequently, 70 studies were excluded because they did not meet the objective of our review, and after a second evaluation round by independent reviewers, at least 12 studies were included and discussed (Table 1).”

#3 In Figure 1, it is unclear what “qualitative synthesis” means. Please clarify.

#R3 The sentence was replaced by: ‘Studies analyzed in the manuscript’

#4 On lines 172–174, the explanations of the UPSIT, SST, and B-SIT is unclear and confusing. Please clarify and elaborate on what each test is comprised of and how they are used.

#R4 the explanations of smell test were elaborated:

-“UPSIT is a scratch and sniff microencapsulated odorant strips (40 items) which provides an indication of anosmia. After scratching with a pencil tip, the subject samples the smell and is asked to match it with one of four given choices. A response is required even when no odor is perceived. The test score consists in the total number of correctly identified items. The interpretation of a given subject’s test score is made by its comparison with age and gender-matched population norms provided in the test manual.”

-“SST-12 or 16 olfactory screening test, consisting in a smell identification pen-like in which there are four options words depicting scent objects, in the form of a 4-min identification test allowing to detect anosmia and hyposmia [39,40]. It can also be used laterally, one nostril tested independently of the other.”

-“B-SIT is a quick disposable screening test where the patient is asked to identify 12 odors contained in a microcapsule fixed on 12 strips of paper. This microcapsule is broken and scratched on with a pencil. The test is administered to the patient, who choose from four possible answers to identify the smell. A score under 9 points suggest hyposmia, whereas a score under 4 correspond to anosmia. The test is performed in 5 minutes, and both nasal nostrils are tested simultaneously.”

#5 On lines 184–187, it is stated that “…the UPSIT and the Sniffin’ Sticks Test have been adapted and validated for use in many languages and cultures…” and they have been normalized for different ages and genders. However, it is also stated that “…for that reason other tests have been introduced.” If the UPSIT and Sniffin’ Sticks Test have been modified for such a broad usage, why were other tests developed and why would the seemingly universal and normalized usage of the UPSIT and Sniffin’ Sticks Test be a reason to introduce even more tests? This statement is unclear.

#R5 The sentences were revised and replaced with:Some odors are not universally recognized, and for that reason both the UPSIT and the Sniffin' Sticks Test have been adapted and validated for use in many different languages and cultures, and normative values have been developed for age and gender.”

#6 On line 234, please clarify what “They” refers to. 

#R6 “Brain regions and olfactory function are connected by four associated proteins:…..”

Thank you for appreciating our work and for helping us to make it clearer to the reader with these suggestions.

Round 2

Reviewer 3 Report

The revised manuscript's readability is much improved and is an interesting review on the association of the loss of smell and neurocognitive diseases. I think this adds to the overall scientific community.

I have one minor request before publication: Can the inclusion exclusion section be expanded upon? What were the objectives of this review? Why were the initial 70 studies excluded? What were the requirements to be included? Why were the second 8 studies excluded? and lastly on line 112 "at least 12 studies..." is misleading and the phrasing "at least" should be removed.

Thank you.

Author Response

Reviewer 3) Second round: Comments and Suggestions for Authors

The revised manuscript's readability is much improved and is an interesting review of the association between the loss of smell and neurocognitive diseases. I think this adds to the overall scientific community.

I have one minor request before publication: Can the inclusion exclusion section be expanded upon? What were the objectives of this review? Why were the initial 70 studies excluded? What were the requirements to be included? Why were the second 8 studies excluded? and lastly on line 112 "at least 12 studies..." is misleading and the phrasing "at least" should be removed.

R1) as suggested, we add (red text) and rephrased the reply sentences to suggested questions:

In this manuscript, the inclusion and exclusion criteria were developed after the objectives of this research were finalized but before any research was performed. The objectives of this review were to identify, delineate, and critically evaluate the link between smell and olfactory impairment, as a clinical biomarker of early signs and symptoms of selected neurodegenerative diseases. We conducted a comprehensive literature analysis by searching the "PubMed" database. The principal search terms were ‘olfactory biomarkers’ and ‘cognitive dysfunction’. The filter criteria in the search strategy were: publication date (recent papers published in the last ten years are more informative about a detailed olfactory test used in a clinical study); the language of publication,(only looking at English full articles and not only abstract); clinical studies performed in human subjects and not in animal models; olfactory test evaluation and not imaging/radiological scale; exclusion of many neurodegenerative diseases apart PD, AD, MCI, main psychiatric diseases; exclusion of anosmia caused by Covid-19 infection, and/or other comorbidities for olfactive impairment. The title and abstracts of the identified papers were initially screened and selected by six independent reviewers (IF, GFN, FZ, LC, MGB, and GR) based on their relevance to the review topic. The literature search yielded 90 papers. From a critical recognition of works reporting these aims, more than 70 were excluded (Figure 1). At least, 8 studies were excluded because the similarity in objectives, experimental paradigm, or final interpretation of the data was reported by the researchers. With the intention of avoiding repetitions for readers, only original manuscripts were selected to analyze, and 12 studies were included and discussed (Table 1).
